# VAT-KG: Knowledge-Intensive Multimodal Knowledge Graph Dataset for Retrieval-Augmented Generation

## Abstract

Multimodal Knowledge Graphs (MMKGs), which represent explicit knowledge across multiple modalities, play a pivotal role by complementing the implicit knowledge of Multimodal Large Language Models (MLLMs) and enabling more grounded reasoning via Retrieval Augmented Generation (RAG). However, existing MMKGs are generally limited in scope: they are often constructed by augmenting pre-existing knowledge graphs, which restricts their knowledge, resulting in outdated or incomplete knowledge coverage, and they often support only a narrow range of modalities, such as text and visual information. These limitations restrict applicability to multimodal tasks, particularly as recent MLLMs adopt richer modalities like video and audio. Therefore, we propose the Visual-Audio-Text Knowledge Graph (VAT-KG), the first concept-centric and knowledge-intensive multimodal knowledge graph that covers visual, audio, and text information, where each triplet is linked to multimodal data and enriched with detailed descriptions of concepts. Specifically, our construction pipeline ensures cross-modal knowledge alignment between multimodal data and fine-grained semantics through a series of stringent filtering and alignment steps, enabling the automatic generation of MMKGs from any multimodal dataset. We further introduce a novel multimodal RAG framework that retrieves detailed concept-level knowledge in response to queries from arbitrary modalities. Experiments on question answering tasks across various modalities demonstrate the effectiveness of VAT-KG in supporting MLLMs, highlighting its practical value in unifying and leveraging multimodal knowledge.

## 1 Introduction

Multimodal Knowledge Graphs (MMKGs) integrate heterogeneous data into a unified graph representation, supporting tasks such as retrieval Alberts et al. (2021); Zeng et al. (2023), reasoning Zhang et al. (2023c); Gong et al. (2024), and question answering Zha et al. (2024). Recent works Zha et al. (2024); Lee et al. (2024); Liu et al. (2025) have proposed Retrieval Augmented Generation (RAG) based approaches that leverage explicit knowledge structures from MMKGs to reduce hallucination in Multimodal Large Language Models (MLLMs). As shown in the upper part of Fig. 1, MLLMs often fail to produce an accurate response in the knowledge-intensive scenario requiring fine-grained facts, highlighting the need for external structured knowledge. Moreover, existing MMKG-based RAG methods are limited to narrow modality pairs (*e.g.* image-text), and thus are not well-suited for recent MLLMs designed for joint understanding of video, audio, and text Liu et al. (2023); Zhang et al. (2023a); Tang et al. (2023); Hong et al. (2023).

The limitations of existing MMKGs are illustrated in Fig. 1. As shown in part (a), although some MMKGs Wang et al. (2023) cover multiple modalities, they primarily focus on entity-to-entity connectivity and do not organize knowledge around individual concepts. For example, in the upper part of Fig. 1, simple triplets such as (quokka; RelatedTo; australia) or (quokka; IsA; mammal) would fail to provide detailed knowledge about the concept. In part (b), a few MMKGs Zha et al. (2024); Lee et al. (2023) adopt a concept-centric structure, where entities are enriched with concept-level descriptions capturing their meaning and context. However, they support only two modalities—image and text—which makes them insufficient for recent MLLMs designed to jointly understand video, audio, and text.

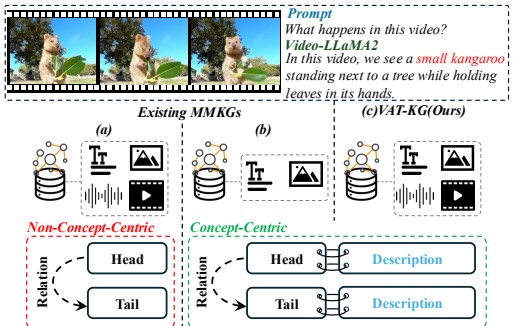

Figure 1: **Comparison with Existing MMKGs.** **(a)** A non-concept-centric case involving various modalities. **(b)** Concept-centric case with limited modalities. **(c)** Our proposed VAT-KG, which is concept-centric and covers four modalities.

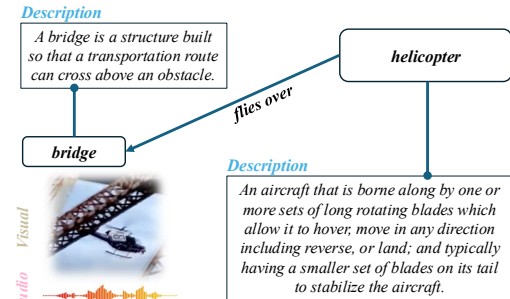

Figure 2: **Example triplet in VAT-KG.** A multimodal triplet from VAT-KG, composed of video, audio, and text. Each head and tail is linked to a detailed concept-level description.

To address these limitations of existing MMKGs, we present the Visual-Audio-Text Knowledge Graph (**VAT-KG**), the first concept-centric and knowledge-intensive multimodal knowledge graph dataset that covers visual, audio, and text, designed to provide explicit cross-modal knowledge to MLLMs. Our VAT-KG construction begins with a multimodal corpus comprising video, audio, and text, and follows a four-step pipeline. First, we perform **(1) Multimodal Alignment Filtering** to assure correspondence between modalities within the multimodal corpus. This step makes it possible to incorporate even those corpora lacking robust multimodal correlations into the construction of VAT-KG. Following alignment, we implement **(2) Knowledge-Intensive Recaptioning** and **(3) Multimodal Triplet Grounding** to extract knowledge-intensive triplets from textual data, leveraging advanced LLM. Finally, a **(4) Cross-Modal Description Alignment** step matches the multimodal concepts within the triplets to detailed descriptions crawled from diverse knowledge bases.

As illustrated in Fig. 2, VAT-KG links each concept to detailed descriptions and organizes its relations as multimodal triplets. This design of VAT-KG plays a pivotal role in multimodal RAG, as it enables the retrieval of detailed descriptions in response to queries from arbitrary modalities. To support this, we additionally introduce a multimodal RAG framework that leverages VAT-KG's concept-centric and multimodal structure to retrieve detailed descriptions in response to queries from arbitrary modalities. Experimental results on question answering tasks across various modalities demonstrate that incorporating VAT-KG into the RAG pipeline leads to substantial and consistent performance gains, significantly boosting the capabilities of MLLMs in handling complex multimodal inputs.

Our contributions are summarized as follows:

- To the best of our knowledge, we present the first knowledge-intensive and concept-centric multimodal knowledge graph covering visual, audio, and text modalities, enriched with detailed descriptions of each concept.
- We introduce an effective pipeline for constructing multimodal knowledge graphs from arbitrary multimodal corpora, ensuring cross-modal alignment between data and fine-grained knowledge through rigorous filtering and alignment steps.
- We propose a multimodal RAG framework that retrieves semantically relevant knowledge in response to queries from arbitrary modalities and refines it using a Retrieval Checker module that filters out misaligned results.
- We demonstrate the effectiveness of VAT-KG with our Multimodal RAG framework, achieving consistent performance gains on Audio QA (AQA), Video QA (VQA), and Audio-Visual QA (AVQA), highlighting its utility in real-world multimodal scenarios.

## 2 RELATED WORKS

### 2.1 MULTIMODAL KNOWLEDGE GRAPH CONSTRUCTION

Multimodal knowledge graphs (MMKGs) have gained attention for semantically bridging text, image, audio, and video data to support cognitive understanding across modalities. A notable example is

Table 1: Overview of notable Multimodal Knowledge Graph (MMKG) constructions.

| Dataset | Year | Modality | | | | Concept centric | Downstream task | Data source |
| | | Text | Image | Audio | Video | | | |
|---|---|---|---|---|---|---|---|---|
| IMGpedia Ferrada et al. (2017) | 2017 | ✓ | ✓ | ✗ | ✗ | ✗ | Link-prediction | Wikimedia Commons, DBpedia |
| ImageGraph Liu et al. (2017) | 2017 | ✓ | ✓ | ✗ | ✗ | ✗ | Local Ranking | FB15k |
| MMKG Liu et al. (2019) | 2019 | ✓ | ✓ | ✗ | ✗ | ✗ | Link-prediction, Reasoning | FB15k, DB15k, YAGO15k, Search Engine |
| Richpedia Wang et al. (2020) | 2020 | ✓ | ✓ | ✗ | ✗ | ✗ | Retrieval | Wikidata, Wikimedia, Search Engine |
| VisualSem Alberts et al. (2021) | 2020 | ✓ | ✓ | ✗ | ✗ | ✗ | Retrieval | BabelNet |
| MarKG Zhang et al. (2022) | 2023 | ✓ | ✓ | ✗ | ✗ | ✗ | Link-prediction, Reasoning | Wikipedia, Search Engine |
| AspectMMKG Zhang et al. (2023b) | 2023 | ✓ | ✓ | ✗ | ✗ | ✗ | Entity aspect linking | Wikipedia, Search Engine |
| VTKG Lee et al. (2023) | 2023 | ✓ | ✓ | ✗ | ✗ | ✓ | Link-prediction | ConceptNet, WordNet |
| TIVA-KG Wang et al. (2023) | 2023 | ✓ | ✓ | ✓ | ✓ | ✗ | Link-prediction | Wikipedia, Search Engine |
| UKnow Gong et al. (2024) | 2024 | ✓ | ✓ | ✗ | ✗ | ✗ | Reasoning, Retrieval | Wikipedia, News |
| M²ConceptBase Zha et al. (2024) | 2024 | ✓ | ✓ | ✗ | ✗ | ✓ | VQA | Image-text Corpora, Encyclopedia, LLM |
| VAT-KG | 2025 | ✓ | ✓ | ✓ | ✓ | ✓ | AQA, VQA, AVQA | Video-Audio-Text Corpora, Encyclopedia, LLM |

MMKG Liu et al. (2019), which aligns three knowledge graphs across FB15K Bordes et al. (2013), DBpedia Auer et al. (2007), and YAGO Suchanek et al. (2007), enriching each entity with images and numerical attributes to support multi-relational link prediction and entity matching. As summarized in Tab. 1, recent studies have effectively leveraged visual information for multimodal reasoning, enriching semantic representations and broadening knowledge coverage. Richpedia Wang et al. (2020) and VisualSem Alberts et al. (2021) aim to establish high-quality multimodal knowledge graphs by retrieving and filtering relevant visual and textual data. In addition, MarKG Zhang et al. (2022) transfers MMKGs to analogical reasoning by constructing a multimodal benchmark grounded in cross-modal relational patterns. Despite recent advances, MMKGs fall short of comprehending entities holistically, as they often rely on narrowly scoped perspectives and overlook the diverse attributes of entities. To tackle these problems, AspectMMKG Zhang et al. (2023b) designs the first aspect-aware multimodal KG, linking entities to multiple aspect-specific images to capture fine-grained semantic facets of each entity. VTKG Lee et al. (2023) attaches visual context to both entities and relational triples while providing textual descriptions for each entity and relation. TIVA-KG Wang et al. (2023) uniquely grounds multimodal data (*i.e.*, text, image, video, audio) at the triple level to capture complex relations beyond simple entity-media links. UKnow Gong et al. (2024) integrates entity- and concept-level knowledge from images and text under one framework, enabling unified cross-modal reasoning beyond image-text pair corpora. Recently, M²ConceptBase Zha et al. (2024) is to provide a concept-centric multimodal knowledge base designed for fine-grained alignment between visual content and semantic concepts.

## 2.2 MULTIMODAL LARGE LANGUAGE MODEL

Recent advances in Multimodal Large Language Models (MLLMs) have proven that general-purpose reasoning capabilities can be grounded beyond language, leveraging visual, auditory, and spatial modalities in various open-ended tasks (*e.g.*, Visual Question Answering, Image Captioning, among others). Early models such as Flamingo Alayrac et al. (2022) and BLIP-2 Li et al. (2023b) pioneered connecting frozen LLMs with vision encoders, laying the groundwork for multimodal reasoning. Scaling efforts from LLaVA Liu et al. (2023) to GPT-4V Achiam et al. (2023) further advanced generalist visual understanding through large-scale pre-training and instruction tuning. Beyond vision, SALMONN Tang et al. (2023) incorporates audio inputs into an LLM by integrating speech and sound encoders, allowing the model to handle speech recognition, auditory question answering, and music understanding within one framework. For audio–visual understanding, Video-LLaMA Zhang et al. (2023a) and its successor Video-LLaMA2 Cheng et al. (2024) incorporated temporal vision features and audio cues, using a Video-Q-Former together with an ImageBind-based Girdhar et al. (2023) Audio-Q-Former. Most recently, Qwen2.5-Omni Xu et al. (2025) exemplifies strong omni-input reasoning, showcasing the shift toward models capable of seamlessly handling diverse modalities.

## 3 DATASET CONSTRUCTION METHOD

In this section, we provide a detailed explanation of how VAT-KG is constructed, along with key statistics highlighting its quality. Sec. 3.1 outlines our four-step construction pipeline, while Sec. 3.2 presents detailed statistics. The overall construction process is illustrated in Fig. 3. Additional implementation details for VAT-KG construction are provided in Appendix C.1.

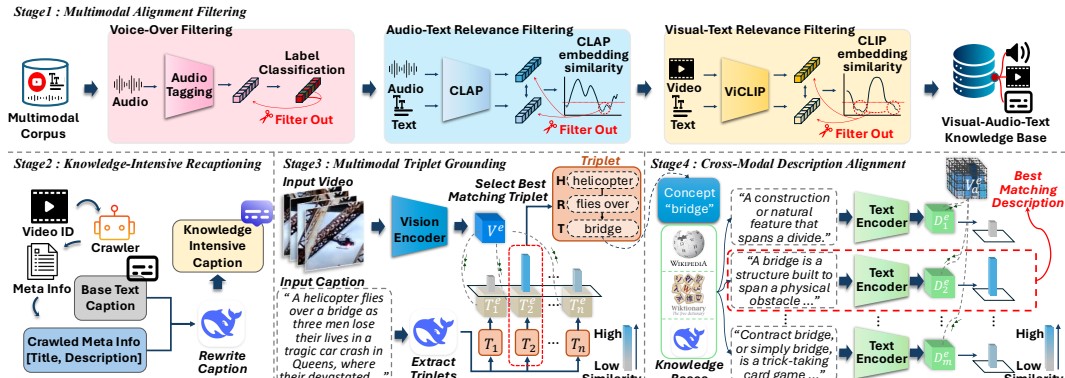

Figure 3: **An overview of the VAT-KG construction pipeline.** The construction of VAT-KG involves four stages: (1) Multimodal Alignment Filtering, ensuring correlation across modalities; (2) Knowledge-Intensive Recaptioning, which transforms base text into rich, knowledge-intensive caption based on meta information; (3) Multimodal Triplet Grounding, which aligns triplets with corresponding multimodal context; (4) Cross-Modal Description Alignment, which retrieves fine-grained descriptions from external knowledge bases and matches them to each multimodal triplet.

## 3.1 VAT-KG CONSTRUCTION PIPELINE

**Stage 1: Multimodal Alignment Filtering** Inspired by Zha et al. (2024), our multimodal knowledge graph construction begins with a multimodal corpus comprising video, audio, and text (*caption or label*). Given that our MMKG is designed such that each triplet is represented by a set of multimodal data, it is essential that all modalities are strictly aligned to a shared semantic context. Therefore, our construction of VAT-KG must begin with multimodal corpus with strong alignment between audio, visual, and textual modalities. However, while most existing multimodal datasets are constructed as video-text datasets Wang et al.; Nan et al. (2025); Bain et al. (2021), they often overlook audio-visual correlations. Although datasets used for audio classification Chen et al. (2020); Gemmeke et al. (2017) or audio-visual tasks Sun et al. (2024) typically exhibit strong correlations between audio and visual modalities, they often suffer from noise and overlook visual-textual alignment. Therefore, we designed a Multimodal Alignment Filtering stage that measures the relevance between text captions and other modalities, enabling the selection of well-aligned multimodal data. Following a similar approach to Sun et al. (2024), we first perform **Voice-over Filtering**. We use a pretrained audio tagging model Schmid et al. (2023) to predict tags of sound, and if the top-5 predictions contain both *speech* and *audio* labels, we regard the sample as an explanatory video with background music and remove it. Such data are excluded because the auditory information does not match the visual and textual modalities. Next, we perform **Audio-Text Relevance Filtering** using the off-the-shelf CLAP Wu et al. (2023) model. We extract audio and text features using CLAP and compute their cosine similarity, filtering out samples with low scores. This step removes cases where the audio does not align with the overall multimodal context–such as background music or sound effects in edited videos. Finally, to ensure consistency between video and text, we perform **Video-Text Relevance Filtering** leveraging ViCLIP Wang et al.. Following a similar approach to audio-text relevance filtering, we extract video and text features with ViCLIP and compute their cosine similarity. We then remove the bottom 10% of multimodal samples with the lowest similarity scores. After a rigorous filtering process, we extract the center frame of each video, which serves as the visual representation in VAT-KG.

**Stage 2: Knowledge-Intensive Recaptioning** Given curated multimodal corpora with strong cross-modal correlations, we expect the corresponding textual data to exhibit knowledge-intensive properties. Since we extract triplets from textual data using an LLM, the text must contain fine-grained details to construct knowledge-intensive MMKG. However, most existing large-scale multimodal datasets Wang et al.; Xiong et al. (2024b); Chen et al. (2024) rely on automatic captioning using MLLMs. While the generated text captions sufficiently capture the overall context, they are limited by the knowledge of the MLLMs and often omit fine-grained details, such as object-level semantics. To enrich the base text captions with knowledge-intensive content, we design a Knowledge-Intensive Recaptioning stage that leverages additional metadata (*video title and description*) crawled from

YouTube. During this stage, data for which such metadata could not be retrieved due to privacy restrictions or limited availability are excluded. We reconstruct the original textual data using a cutting-edge LLM Guo et al. (2025), which takes both the original text and the retrieved metadata as input and generates refined captions, guided by a carefully designed prompt. Through this process, the reconstructed text captions go beyond describing the overall context and begin to incorporate background knowledge and fine-grained semantic details. As a result, the textual data becomes significantly more knowledge-intensive and informative than the original one.

**Stage 3: Multimodal Triplet Grounding** Building on the knowledge-intensive captions from the previous stage, we designed the third stage leveraging LLM Guo et al. (2025) and ViCLIP Wang et al. to extract and associate fine-grained triplets with corresponding multimodal data. Extracting KG triplets from text using LLMs has been an active area of research, with studies demonstrating the effectiveness and applicability of this approach for automatic KG construction Pan et al. (2024); Zhang & Soh (2024). Inspired by these previous works, we utilize a cutting-edge LLM to generate multiple candidate triplets from each caption. In addition, motivated by prior work Brown et al. (2020) showing in-context learning, we thoughtfully design a prompt for triplet grounding that includes in-context examples. As the text caption mostly contains multiple knowledge elements, the LLM typically generates several candidate triplets for each caption. To select the triplet that best reflects the corresponding multimodal data, we leverage ViCLIP. Given $n$ candidate triplets, we convert each triplet into a natural language sentence denoted as $(T_1, T_2, ..., T_n)$ by concatenating its head, relation, and tail. We then encode these sentences using ViCLIP's text encoder to obtain their corresponding embeddings $(T_1^e, T_2^e, ..., T_n^e)$. To determine which triplet best represents the multimodal input, we select the triplet with the highest inner product between its text embeddings and the corresponding video embedding $V^e$. This stage yields knowledge-intensive triplets that are best aligned with the multimodal data.

**Stage 4: Cross-Modal Description Alignment** In the final stage, to make our knowledge graph concept-centric, which provides detailed information for each associated concept, we crawl rich textual descriptions from multiple knowledge bases and link them to their corresponding multimodal concepts. For knowledge bases, we utilize Wikipedia, Wiktionary, and LLM (*e.g.* DeepSeek-R1) to mine detailed encyclopedic descriptions. Since the meaning of a single text concept may vary significantly depending on the multimodal context (*e.g.* concept *goal* may refer to a personal ambition or a soccer score), we collect a maximum of 5 candidate descriptions per concept from external knowledge bases. As Wikipedia and Wiktionary are manually curated with high reliability, they are prioritized as primary sources during the crawling process. While two knowledge bases offer high-quality descriptions, they do not cover all textual concepts. To compensate for this limitation, we leverage the implicit knowledge of LLM to supplement missing descriptions, ensuring that each concept contains a detailed textual explanation. After obtaining detailed descriptions for all concepts, we select the description that best matches the semantic content for each multimodal data. Following a similar approach to Zha et al. (2024), we first compute an attention-weighted video that highlights regions relevant to the given concept. Among the triplet components, the head and tail are concepts that can be visually grounded in the data. By following the formulation of Chefer et al. (2021), we can localize their corresponding regions within the video. Unlike prior work Zha et al. (2024) that matches attention-weighted images with textual descriptions, we employ ViCLIP Wang et al. to compute attention-weighted video ($V_a$), enabling alignment between the full video context and corresponding concepts. After deriving the attention-weighted video, we feed each frame into ViCLIP to extract video embeddings ($V_a^e$), and compute their similarity with the text embeddings of candidate descriptions ($D_1^e, D_2^e, ..., D_m^e$). The description with the highest similarity is selected as the one that best matches the multimodal context.

## 3.2 DATASET SPECIFICATIONS AND STATISTICS

In Tab. 1, we compare existing MMKGs with our proposed VAT-KG. Unlike other MMKGs, VAT-KG is the only knowledge graph that incorporates all four modalities(text, image, audio, video) and contains triplets enriched with concept-level descriptions at the same time. This enables VAT-KG to be applied to multiple downstream tasks across different modalities.

We leverage four datasets–InternVid-FLT Wang et al., AudioCaps Kim et al. (2019), AVQA Yang et al. (2022), and VALOR-32k Liu et al. (2024)–as the source multimodal corpora for constructing VAT-KG. Due to the high computational cost arising from the large scalability of InternVid, we

Table 2: Data sample counts at each stage of the filtering process.

| Principle | InternVid-FLT(10%) | AudioCaps | AVQA | VALOR-32k | Total |
|---|---|---|---|---|---|
| *Original* | 1,000,000 | 93,726 | 40,150 | 28,823 | 1,162,699 |
| Audio Tagging | 389,965 | 86,578 | 36,144 | 22,521 | 535,208 |
| Audio-Text | 15,490 | 77,964 | 27,864 | 16,945 | 136,568 |
| Video-Text | 13,941 | 70,167 | 25,077 | 15,250 | 124,295 |
| Final | 10,762 | 59,808 | 24,999 | 15,217 | **110,786** |

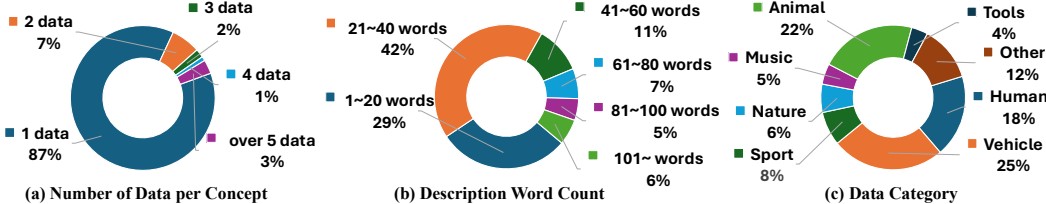

(a) Number of Data per Concept   (b) Description Word Count   (c) Data Category

Figure 4: **Statistics of VAT-KG.** **(a)** VAT-KG contains diverse concepts that are represented through varied multimodal data. **(b)** Concept-level descriptions linked to VAT-KG concepts are sufficiently comprehensive and informative. **(c)** VAT-KG ensures diversity across categories.

sample 10% of the InternVid-FLT for VAT-KG construction. For all datasets, only the training split is used for construction. Tab. 2 presents the number of data filtered at each step of the Multimodal Alignment Filtering process. Unlike the other datasets, InternVid-FLT shows a larger reduction in data during the filtering process. This is expected, as InternVid is originally designed for video-text tasks and does not explicitly ensure strong audio associations, which naturally results in more samples being filtered in our audio-related alignment filtering steps. A carefully designed filtering process retains 124,295 high-quality instances with strong cross-modal correlation from an initial 1,162,699. Following the exclusion of samples without meta information in the Knowledge-Intensive Recaption stage, we finally use **110,786** multimodal data to construct VAT-KG. After completing the remaining construction stages, we obtain a total of **102,203** unique concepts and **110,786** triplets, each paired with corresponding multimodal data. We additionally validate the quality of VAT-KG through comprehensive human evaluation, with details provided in Appendix D.1.

Fig. 4 presents overall statistics of our VAT-KG, including the number of multimodal data per concept, description length distribution, and category distribution. Following a similar approach to Xiong et al. (2024a), we use BART Lewis et al. (2020) for categorization by first converting each triplet into a natural language sentence via concatenation and then providing it as input. We adopt the category taxonomy used in Geng et al. (2023).

## 4 MULTIMODAL RAG FRAMEWORK

Given that existing RAG frameworks Jian et al. (2024); Jeong et al. (2025) do not jointly consider video, audio, and text, we design a multimodal RAG framework tailored to the modality coverage of VAT-KG, supporting AQA, VQA, and AVQA tasks. As illustrated in Fig. 5, our model retrieves concept-level knowledge from VAT-KG in response to queries from arbitrary modalities and serves fine-grained, contextually aligned knowledge to MLLMs. Our framework is composed of three main components. Additional implementation details are provided in Appendix C.2.

### 4.1 MODALITY-AGNOSTIC RETRIEVAL

Given a query from an arbitrary modality, we first encode the input data using appropriate multimodal foundation models (*e.g.* ViCLIP for video and CLAP for audio) to obtain the corresponding query embedding ($V^e$ or $A^e$). We then retrieve up to five relevant triplets by comparing the query embedding against triplet embeddings computed from linked data of the same modality as the query. Retrieval is restricted to candidates within a predefined L2 distance threshold in the VAT-KG embedding space. The embedding space of VAT-KG is indexed with FAISS Johnson et al. (2019) for fast retrieval. For

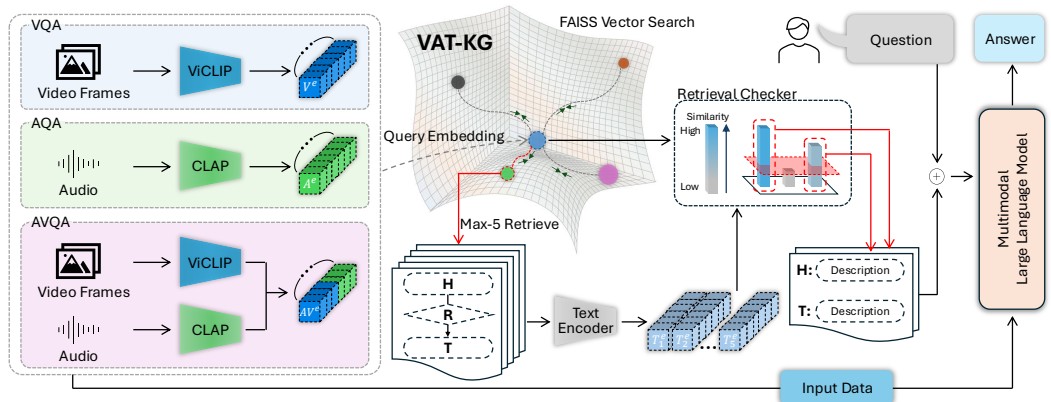

Figure 5: **An overview of Multimodal RAG Framework.** Given a query from any modality (audio, video, or text), **(1)** Modality-Agnostic Retrieval retrieves up to five semantically relevant triplets from VAT-KG based on embedding similarity; **(2)** Retrieval Checker filters out misaligned triplets by using the text encoder of the same multimodal foundation model; **(3)** Augmented Generation with MLLMs that support audio-visual understanding.

AVQA, which requires reasoning over both audio and video modalities, we simply concatenate the audio and video embeddings to form a joint query embedding ($AV^e$). Similarity-based retrieval is then performed in the VAT-KG embedding space using the same distance metric as used for single-modality queries.

## 4.2 RETRIEVAL CHECKER

Although we retrieve semantically relevant triplets by performing a similarity-based search using multimodal foundation models, these models often fail to capture fine-grained details despite capturing overall context well. As a result, the retrieved triplet may not precisely match the intent or specific semantics of the query. Inspired by recent re-ranking strategies Abootorabi et al. (2025); Mortaheb et al. (2025) in multimodal RAG, we design a Retrieval Checker module to verify whether the retrieved triplets are semantically aligned with the input query. First, we convert each triplet into a natural sentence by simply concatenating its head, relation, and tail. Then, these sentences are encoded into text embeddings ($T_1^e, T_2^e...T_5^e$) using the text encoder of the same multimodal foundation model used during retrieval, in order to place them in the same embedding space as the query for comparison. Next, we measure the similarity between the query embedding and the text embeddings. If the similarity falls below a predefined threshold, indicating a mismatch with the query context, the triplet is discarded. This double-checking mechanism ensures more precise retrieval by filtering out triplets with subtle semantic mismatches, thereby mitigating potential hallucinations from loosely relevant retrievals.

## 4.3 AUGMENTED GENERATION WITH MLLMs

As our Multimodal RAG baseline model supports audio-visual retrieval, we adopt MLLMs that are designed to jointly understand both audio and video inputs. Following the retrieval of query-relevant triplets from VAT-KG, we utilize the descriptions linked with their head and tail concepts. We feed the input question along with the head and tail, as well as their corresponding descriptions, into MLLMs, enabling a multimodal RAG process that incorporates aligned concept-level knowledge.

## 5 EXPERIMENT

In this section, we evaluate the effectiveness of VAT-KG using our multimodal RAG framework. Sec. 5.1 describes the experimental setup, while Sec. 5.2 presents the performance of our model across various Question-Answering tasks. Additional ablation studies and extended experimental results are provided in Appendix E.

Table 3: **Overall performance.** We report M.J. (Model-as-Judge) Wang et al. (2025) scores; higher is better. VAT-KG yields the highest performance improvements, highlighted in **bold**.

| Method | Knowledge Graph | Audio QA | Video QA | Audio-Visual QA | |
|---|---|---|---|---|---|
| | | AudioCaps-QA | VCGPT | AVQA | VALOR |
| VideoLLaMA2 | None | 43.13 | 39.09 | 93.19 | 25.66 |
| VideoLLaMA2 | Wikidata | 43.58 | 38.58 | 92.70 | 26.43 |
| VideoLLaMA2 | VTKG | 43.02 | 38.88 | 90.52 | 25.92 |
| VideoLLaMA2 | M$^2$ConceptBase | 42.19 | 39.31 | 92.92 | 25.93 |
| VideoLLaMA2 | VAT-KG | **44.60** | **39.42** | **93.28** | **28.30** |
| Qwen2.5-Omni | None | 49.00 | 42.21 | 93.05 | 32.42 |
| Qwen2.5-Omni | Wikidata | 49.78 | 40.82 | 92.90 | 30.28 |
| Qwen2.5-Omni | VTKG | 48.95 | 42.96 | 92.67 | 32.70 |
| Qwen2.5-Omni | M$^2$ConceptBase | 49.78 | 42.78 | 92.28 | 32.31 |
| Qwen2.5-Omni | VAT-KG | **51.30** | **43.50** | **93.07** | **35.44** |

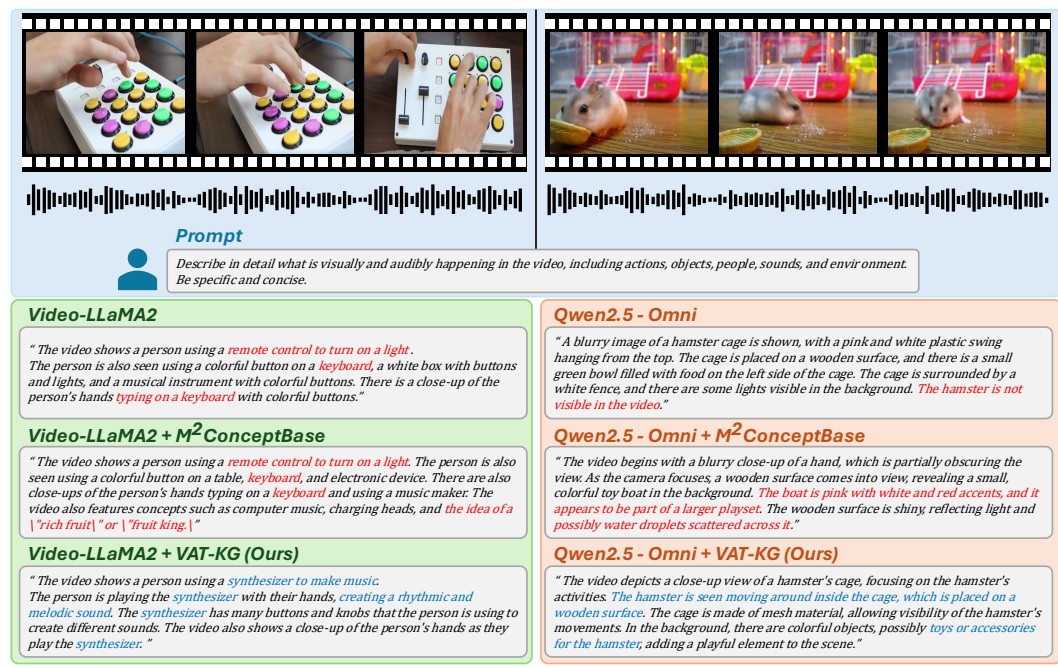

Figure 6: **Qualitative comparison on AVQA task.** Incorrect responses are highlighted in red, while correct answers are shown in blue.

## 5.1 EXPERIMENTAL SETTINGS

**Datasets** We adopt a diverse set of QA benchmarks covering audio, visual, and audio-visual modalities to comprehensively evaluate the performance of our model across various modalities. For the AQA task, we evaluate our model on the AudioCaps-QA Wang et al. (2025). For the VQA task, we adopt the VideoChatGPT (VCGPT) benchmark Maaz et al. (2024). For the AVQA task, we select AVQA Yang et al. (2022) and VALOR Liu et al. (2024) for benchmark. While VALOR was originally designed for audio-visual captioning, we adopt it for evaluating the AVQA task, as in prior work Ye et al. (2024). For each benchmark except VCGPT, we use the subgraph of VAT-KG that is built upon the training set of the corresponding dataset to better match the knowledge domain with the benchmark. Only for VCGPT, we employ a subgraph of VAT-KG constructed from 10% of the InternVid-FLT dataset for multimodal RAG.

**Baseline MLLMs** For the baseline MLLMs, we employ VideoLLaMA2 Cheng et al. (2024) and Qwen2.5-Omni Xu et al. (2025), both of which are advanced models capable of jointly processing audio and visual inputs. These models have demonstrated strong performance in understanding multimodal data simultaneously across visual, auditory, and textual modalities. We further adopt commercial MLLMs (*e.g.* GPT-4o, Gemini-2.5); results are reported in Appendix E.2.

**Knowledge Graphs Comparison** To thoroughly examine the benefits that knowledge from VAT-KG brings to MLLMs, we conduct a comprehensive comparison against three knowledge graphs, namely

Wikidata5M Wang et al. (2021), VTKG Lee et al. (2023), and M$^2$ConceptBase Zha et al. (2024). A detailed description of each knowledge graph and its integration into the multimodal RAG framework are provided in the Appendix C.3.

**Evaluation Metrics** Given that the majority of our benchmarks involve open-ended QA, we employ the Model-as-Judge (M.J.) approach as our evaluation metric. To avoid high cost and inconsistency caused by version changes in cloud-based commercial models like GPT-4 Achiam et al. (2023), we adopt the open-source M.J. methods proposed in recent work Wang et al. (2025). In addition, we complement this automatic evaluation with human assessment, as described in Appendix D.2.

## 5.2 EXPERIMENTAL RESULTS

We evaluate our Multimodal RAG baseline by comparing the performance of baseline MLLMs with and without its integration, and also report comparisons with diverse knowledge graphs.

**Quantitative Results** We summarize the overall performance on Audio QA (AQA), Video QA (VQA), and Audio-Visual QA (AVQA) tasks in Tab. 3. Our multimodal RAG framework, when leveraging VAT-KG, consistently yields the highest performance across all tasks, significantly improving the baseline capabilities of both VideoLLaMA2 Cheng et al. (2024) and Qwen2.5-Omni Xu et al. (2025). These improvements stem from VAT-KG's capability to provide knowledge directly linked to multimodal content, enabling retrieval that is more relevant and contextually aligned with arbitrary multimodal queries. In contrast, knowledge retrieved from other knowledge graphs provides only marginal gains—or in some cases even degrades performance—highlighting their limited utility in multimodal tasks.

The advantage of VAT-KG is most pronounced in the AVQA task, where queries require knowledge retrieval considering both spatial and temporal features of audio and video inputs. In such settings, existing knowledge graphs often lead to performance degradation, whereas VAT-KG achieves notable and consistent improvements. These results underscore that existing large-scale KGs and MMKGs remain insufficient for real-world scenarios requiring both audio and visual understanding, as their knowledge grounding is restricted primarily to image or text modalities.

**Qualitative Results** Fig. 6 shows qualitative comparison results on the AVQA task. Without RAG, both VideoLLaMA2 and Qwen2.5-Omni fail to generate appropriate responses, often misinterpreting the scene or hallucinating irrelevant content. For instance, VideoLLaMA2 fails to describe the musical instrument shown in the video, while Qwen2.5-Omni entirely misses the object depicted in the scene. When leveraging M$^2$ConceptBase, the retrieved knowledge is often misaligned with the context of queries, leading to unrelated responses and reinforcing hallucination.

In contrast, with VAT-KG, both models benefit from access to fine-grained, contextually aligned knowledge grounded in both video and audio. For example, in the case of VideoLLaMA2, knowledge retrieved from VAT-KG enables the model to accurately identify the object as a synthesizer producing music. For Qwen2.5-Omni, the knowledge retrieved from VAT-KG helps the model correctly interpret the scene, capturing the contextual status of the object more faithfully. These results demonstrate the effectiveness of VAT-KG in providing aligned, concept-centric knowledge across both visual and auditory modalities, ultimately reducing hallucination and improving response accuracy.

## 6 CONCLUSION

In this work, we address the challenge that advanced MLLMs often hallucinate while lacking MMKGs with diverse modalities and fine-grained concept-level knowledge. We propose VAT-KG, the first knowledge-intensive and concept-centric MMKG that comprehensively covers visual, audio, and text modalities, constructed from carefully filtered datasets with strong cross-modal alignment. We also introduce a multimodal RAG framework that retrieves and delivers highly relevant, fine-grained knowledge in response to queries across different modalities. Experimental results on downstream tasks underscore the practical value of VAT-KG, demonstrating its ability to serve diverse and detailed knowledge. We believe that our VAT-KG and multimodal RAG framework will serve as valuable resources for future work on mitigating hallucinations in multimodal large language models and constructing aligned multimodal knowledge.

ETHICS STATEMENT

All datasets used in this work are publicly available multimodal resources that were originally released as part of published academic work. Each dataset is distributed under permissive open-source licenses (e.g., CC-BY-4.0, MIT), and we adhered to the respective licensing terms. During the construction of VAT-KG, we exclude any YouTube videos that have been deleted, set to private, or otherwise restricted, ensuring that only content explicitly made public by the uploader is included. This procedure mitigates potential privacy risks and reduces the likelihood of unintentional user data exposure. Following established practive in prior multimodal datasets, we release only YouTube video IDs rather than raw multimedia content, in compliance with YouTube's Terms of Service. To further safeguard responsible use, we conducted a content safety analysis combining Google Cloud's SafeSearch API with manual inspection, which confirmed the absence of violent, adult, or otherwise inappropriate material.

REPRODUCIBILITY STATEMENT

To ensure reproducibility, we release all resources associated with this work, including the dataset construction pipeline, the multimodal RAG framework, and the curated VAT-KG dataset, through our public HuggingFace repository. Implementation details are provided in the Appendix (see Section C). In addition, we include a detailed description of the open-source datasets and codes used in our work in the Appendix (See Section A). Together, these resources allow independent researchers to reproduce our results and extend our framework with minimal effort.

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

APPENDIX

In this appendix, we provide supplementary details and analyses to support the main paper. We begin by outlining the usage of open-source datasets and codes (Appendix A), followed by a discussion of the limitations of our approach (Appendix B). We then present additional implementation details for both the VAT-KG construction pipeline and the multimodal RAG framework (Appendix C), describe the setup and results of our human evaluation (Appendix D), and conclude with further experimental analyses and ablations (Appendix E).

## A    USAGE OF OPEN SOURCE DATASETS AND CODES

We release the full implementation of the VAT-KG construction pipeline and the Multimodal RAG framework at `https://huggingface.co/iclr26/VATKG_CODE`. The VAT-KG dataset is publicly available at `https://huggingface.co/datasets/iclr26/vat_kg`. For more details, please visit our project page at `https://iclrvatkg.github.io/`.

**Licenses** The dataset is provided under the CC BY-NC 4.0 license and is intended solely for non-commercial research and educational use.

## B    LIMITATIONS

A limitation of our work is that our VAT-KG construction must be built upon multimodal corpora containing video, audio, and text captions. As a result, the diversity of VAT-KG is inherently dependent on the underlying multimodal datasets used during construction. VAT-KG has a smaller data scale compared to other multimodal datasets. Nevertheless, all data included in VAT-KG undergo a rigorous filtering process, ensuring high multimodal correlation and overall quality. We plan to expand the scale of the dataset in future versions to improve coverage and generalization.

## C    IMPLEMENTATION DETAILS

### C.1    VAT-KG CONSTRUCTION

In this section, we provide additional implementation details for the VAT-KG construction process.

#### C.1.1    DETAILS OF CONSTRUCTION PIPELINE

During Audio-Text Relevance filtering in Stage 1, we filter out data samples whose cosine similarity between CLAP Wu et al. (2023) audio embeddings and text embeddings falls below 0.2. In Stages 2 and 3 (Knowledge-Intensive Recaptioning and Multimodal Triplet Grounding), we use DeepSeek-R1-Distill-Llama-70B Guo et al. (2025), the largest variant among the available DeepSeek-R1 distilled models, to avoid the high cost of commercial APIs (*e.g.* GPT-4), which are commonly used in other KG construction pipelines Zha et al. (2024); Zhang & Soh (2024).

The LLM prompt used in Knowledge-Intensive Recaptioning is shown in Fig. 7. Fig. 8 presents the prompt employed for Multimodal Triplet Grounding, which includes in-context examples to guide triplet extraction.

For description crawling in the Cross-Modal Description Alignment stage, we utilize DeepSeek-R1-Distill-Llama-8B as the knowledge base. Since it is used as a sub-knowledge base supporting other knowledge bases, we choose a lighter LLM to reduce inference overhead. The LLM is prompted with the template shown in Fig. 9 to generate candidate descriptions for the given concept. All construction stages use a single H100 GPU (80GB), except Stage 2 and Stage 3, which use two GPUs.

#### C.1.2    DATASETS FOR VAT-KG CONSTRUCTION

**InternVid-FLT** InternVid-FLT Wang et al. is a large-scale multimodal dataset consisting of 10 million videos and their corresponding rich text captions with high video-text correspondence. The video data is scraped from YouTube across diverse content categories, while the captions are synthesized by an LLM from coarse to fine-grained levels.

## Prompt Details for Knowledge-Intensive Recaptioning

You are a Video understanding expert specializing in refining video descriptions.
Given information about one video, your task is to rewrite its caption using additional metadata.
Here is the provided data:
- **Video Caption**: {video_caption}
- **YouTube Title**: {youtube_title}
- **YouTube Description**: {youtube_description}
Your goal is to create a new caption that combines the information from the video caption, YouTube title, and YouTube description. The new caption should be a single paragraph with full sentences in English and should be coherent and informative.

- Do not simply concatenate the inputs. Instead, synthesize them naturally.
- Verify the relevance of the YouTube title and description before using them. If they align with the video caption, incorporate useful details; otherwise, rely primarily on the video caption.
- The output must be a complete and well-structured sentence.
- Do not explain your reasoning. Do not include any extra text. Only output the final sentence.

**The final rewritten caption is:**

Figure 7: **Prompt Details for Knowledge-Intensive Recaptioning.** We prompt the LLM with YouTube metadata (*title and description*) to generate knowledge-intensive textual data.

## Prompt Details for Triplet Grounding

You are an expert in extracting structured knowledge from text.
Given a video caption, extract all possible subject-relationship-object triples in the form (h, r, t).

**Instruction**
- Extract multiple (h, r, t) triples if applicable.
- Ensure each triple is meaningful and correctly represents the relationships in the text.
- Format the output as a list of triples.

**Examples**
Caption: "A man is kicking a soccer ball."
Output: (man, kicks, soccer ball)

Caption: "A man is kicking a soccer ball while a dog is running nearby."
Output:
(man, kicks, soccer ball)
(dog, runs, nearby)

Caption: "A chef is cutting vegetables on a wooden board, and a waiter is bringing a dish."
Output:
(chef, cuts, vegetables)
(chef, works_on, wooden board)
(waiter, brings, dish)

Caption: "{video_caption}"
Output Format: (h, r, t), no explanation.
**Output:**

Figure 8: **Prompt Details for Triplet Grounding.** To extract triplets from knowledge-intensive textual data, we provide in-context examples to guide LLM, facilitating accurate triplet grounding.

---

**Prompt Details for Description Crawling from LLM**

**Instruction**

Explain the concept of '{concept}' in a formal and concise English sentence.

**Response**

---

Figure 9: **Prompt Details for Description Crawling from LLM.** This prompt is used to mine concept-level descriptions from the LLM-based knowledge base.

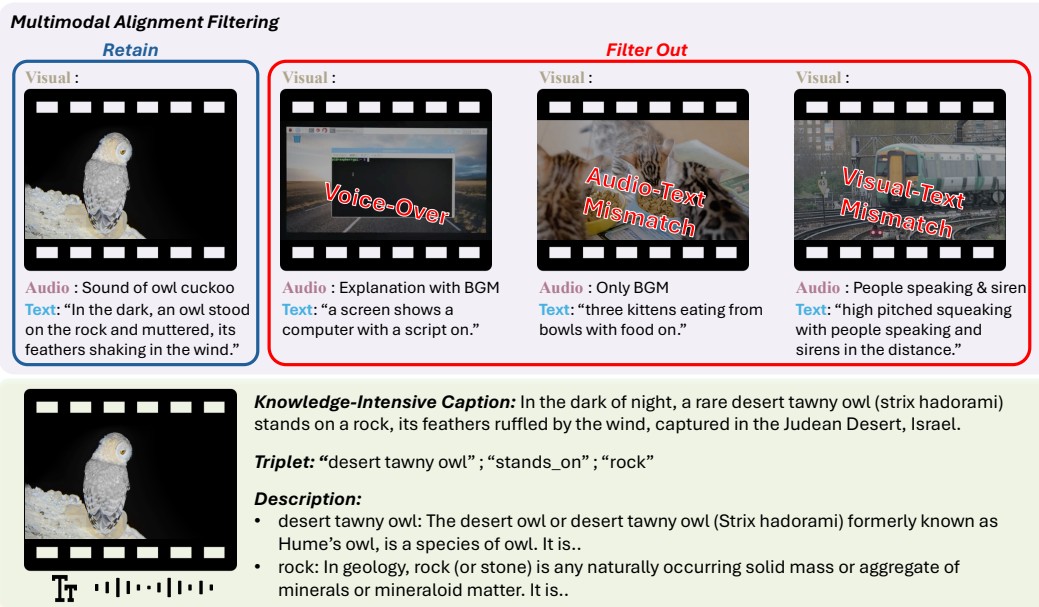

Figure 10: **Qualitative visualization of the VAT-KG pipeline stages.** Given a raw multimodal corpus, our construction pipeline filters out samples with cross-modal mismatches and builds a knowledge-intensive, concept-centric multimodal knowledge graph through the subsequent stages.

**AudioCaps** AudioCaps Kim et al. (2019) is a large-scale audio-text dataset built upon AudioSet Gemmeke et al. (2017), consisting of human-written captions that describe audio events in detail. While AudioCaps was originally developed for the audio captioning task, its underlying dataset, AudioSet, is built upon YouTube crawled videos, which we leverage as the visual modality in VAT-KG.

**AVQA** AVQA Yang et al. (2022) is a question answering dataset designed to reflect real-world scenarios that require both audio and visual modalities for accurate reasoning. Built upon VGG-Sound Chen et al. (2020), which includes 309 class-level text labels, we incorporate these labels as textual inputs during VAT-KG construction.

**VALOR** VALOR Liu et al. (2024) is a multimodal dataset designed for audio-visual-language pretraining, containing rich audio-visual captions annotated by humans. For VAT-KG construction, we utilize VALOR-32k, which offers a more balanced distribution of audio classes compared to the larger VALOR-1M variant.

### C.1.3 VISUALIZATION OF VAT-KG CONSTRUCTION

In Fig. 10, we visualize the intermediate outputs from each stage of VAT-KG construction. Our Multimodal Alignment Filtering stage strictly filters out samples with mismatches among visual, audio, and textual modalities, ensuring that only highly aligned multimodal data are retained for VAT-KG construction. In the second stage, Knowledge-Intensive Recaptioning, the coarse concepts (owl) are transformed into more knowledge-intensive ones (desert tawny owl). The third stage, Multimodal Triplet Grounding, generates candidate triplets and selects one that best aligns with the multimodal context. Finally, in the Cross-Modal Description Alignment stage, we mine and align concept-level descriptions that are semantically consistent with the given multimodal data.

> ## Prompt Details for Multimodal RAG Framework
>
> **Question**
> Hint: You are provided with a set of concepts and their descriptions in the form of key-value pairs.
> These concepts may or may not be directly related to the given video.
> When generating your answer, identify any concepts that appear to be relevant to the video's content,
> and incorporate their corresponding descriptions to enrich and elaborate your response.
> Below are concepts and their descriptions (concept: description).
> {concept_1}:{description_1}
> {concept_2}:{description_2}
> ...
> {concept_N}:{description_N}

Figure 11: **Prompt Details for Multimodal RAG Framework.** Our multimodal RAG framework provides MLLM with $N$ concept-description pairs stringently selected through the retrieval checker.

### C.2 MULTIMODAL RAG FRAMEWORK

In this section, we provide additional implementation details of the multimodal RAG framework. After retrieving $N$ query-relevant triplets and descriptions via modality-agnostic retrieval and the checker, the retrieved knowledge is injected into MLLMs using the prompt template in Fig. 11.

### C.3 KNOWLEDGE GRAPHS FOR COMPARISON

In this section, we provide a detailed description of each knowledge graph used in our comparison, together with their integration into our multimodal RAG framework.

**Wikidata5M** Wikidata5M Wang et al. (2021) is a large-scale knowledge graph derived from Wikidata, comprising about 5 million entities, 20 million triplets, and aligned entity descriptions from Wikipedia. Since it consists solely of textual information, we project its textual data into a shared representation space using a text encoder from multimodal foundation model Wu et al. (2023); Wang et al., enabling retrieval of knowledge relevant to a given multimodal query.

**VTKG** VTKG Lee et al. (2023) is a concept-centric multimodal knowledge graph that aligns visual evidence with textual descriptions for entities and relations. However, it releases only pre-computed image embeddings from a ViT-Base model Wu et al. (2020) without raw multimodal data. Accordingly, for visual queries, knowledge retrieval is performed in the ViT-Base embedding space, and for audio queries we project VTKG's textual data into an audio–text space using a text encoder from CLAP Wu et al. (2023) to perform retrieval.

**M$^2$ConceptBase** M$^2$ConceptBase is our primary baseline: a concept-centric MMKG that provides concept-level descriptions together with raw multimodal data. However, it supports only image–text modalities. Accordingly, retrieval for visual queries is conducted in the shared CLIP Radford et al. (2021) embedding space, while audio queries are handled in the same way as for VTKG.

## D HUMAN EVALUATION

In this section, we report human evaluation results to assess both the quality and the practical utility of VAT-KG. We recruit participants through Amazon Mechanical Turk ama (2005), a widely-used platform for human assessment.

### D.1 HUMAN EVALUATION ON VAT-KG QUALITY

To demonstrate the quality and factual integrity of VAT-KG, we conduct three user studies, each targeting different stages of the construction pipeline.

Specially, we sampled 50 outputs each from Stage 2 (Knowledge-Intensive Recaptioning) and Stage 3 (Multimodal Triplet Grounding). Since both stages employ LLM and may therefore introduce errors, we asked 50 human volunteers to evaluate their factual correctness.

Table 4: **Correctness for Stage 2 & Stage 3.** Human assessment for intermediate outputs.

| Stage | Correctness |
|---|---|
| Knowledge-Intensive Recaptioning | 98.84% |
| Multimodal Triplet Grounding | 98.40% |

Table 5: **Correctness for Final Triplets.**

| Criterion | Correctness |
|---|---|
| Concept Accuracy | 97.46% |
| Content Capture Fidelity | 95.26% |
| Description Alignment | 96.54% |

Table 6: **Human Evaluation on Multimodal QA Benchmarks.** VAT-KG yields consistent improvements across all benchmarks, highlighted in **bold**.

| Method | Knowledge Graph | Audiocaps-QA | VCGPT | AVQA | VALOR |
|---|---|---|---|---|---|
| Qwen2.5-Omni | None | 72.25 | 72.79 | 68.78 | 76.47 |
| Qwen2.5-Omni | VAT-KG | **79.41** | **81.83** | **79.00** | **76.92** |

As shown in the Tab. 4, both stages achieved near-perfect correctness scores, which we attribute to the structured nature of the tasks—i.e., paraphrasing and extraction from grounded text—rather than open-ended generation. This aligns with recent studies Zhang & Soh (2024); Li et al. (2023a); Han et al. (2023) showing the effectiveness and reliability of LLMs as open information extractors.

Additionally, to directly evaluate the final graph output, we randomly sampled 30 triplets from VAT-KG and asked human annotators to assess each triplet along three criteria: (i) Concept Accuracy, measuring whether the each concept in the triplet is presented in the multimodal data; (ii) Content Capture Fidelity, evaluating whether the triplet captures the core semantics of the video/audio content; and (iii) Description Alignment, assessing the attached description are well-aligned with the underlying multimodal signals.

As shown in the Tab. 5, the evaluation results indicate that the triplets satisfy nearly all criteria, offering strong empirical evidence that the factual quality of VAT-KG is well-preserved across its construction stages.

### D.2 HUMAN EVALUATION ON MULTIMODAL QA BENCHMARKS

As the Model-as-Judge (M.J.) evaluation may introduce bias, we complement it with human ratings to ensure the reliability of our experimental results. We randomly sampled 10 QA pairs from each of the four benchmarks used in the main experiments and asked 100 human participants to rate the model-generated answers on a 0-5 scale. The final score is computed by averaging the ratings and multiplying the mean by 20, following the same scoring scheme used in the Model-as-Judge (M.J.) evaluation. As summarized in Tab. 6, across all benchmarks, the multimodal RAG framework equipped with VAT-KG consistently achieved the highest human ratings.

These results provide evidence that the M.J. evaluation is aligned with human judgment and that our method yields consistent improvements across both evaluation protocols.

## E ADDITIONAL EXPERIMENTAL RESULTS

### E.1 ABLATION STUDY

**Modality-Wise Retrieval** To evaluate the utility of VAT-KG's multimodal coverage in our multimodal RAG framework, we conduct an ablation study on modality-wise retrieval. Specifically, we compare four retrieval strategies for the AVQA task: retrieving knowledge from VAT-KG based on audio only, image only (middle frame of the query video), video only, and both audio and video. This allows us to assess the impact of leveraging all available query modalities on RAG performance.

We perform this ablation on the VALOR benchmark, using the query audio, video, or both to retrieve relevant knowledge from the shared embedding space of VAT-KG. For the image-based retrieval condition, we extract the middle frame from each query video to use as an image query.

Table 7: **Ablation study on Retrieval Checker**. Applying the Retrieval Checker yields the best performance across all benchmarks, highlighted in **bold**.

| Method | Retrieval Checker | Audio QA | Video QA | Audio-Visual QA | |
|---|---|---|---|---|---|
| | | AudioCaps-QA | VCGPT | AVQA | VALOR |
| VideoLLaMA2 + VAT-KG | ✗ | 43.90 | 39.14 | 93.04 | 26.02 |
| VideoLLaMA2 + VAT-KG | ✓ | **44.60** | **39.42** | **93.28** | **28.30** |
| Qwen2.5-Omni + VAT-KG | ✗ | 50.69 | 43.48 | 93.08 | 33.25 |
| Qwen2.5-Omni + VAT-KG | ✓ | **51.30** | **43.50** | 93.07 | **35.44** |

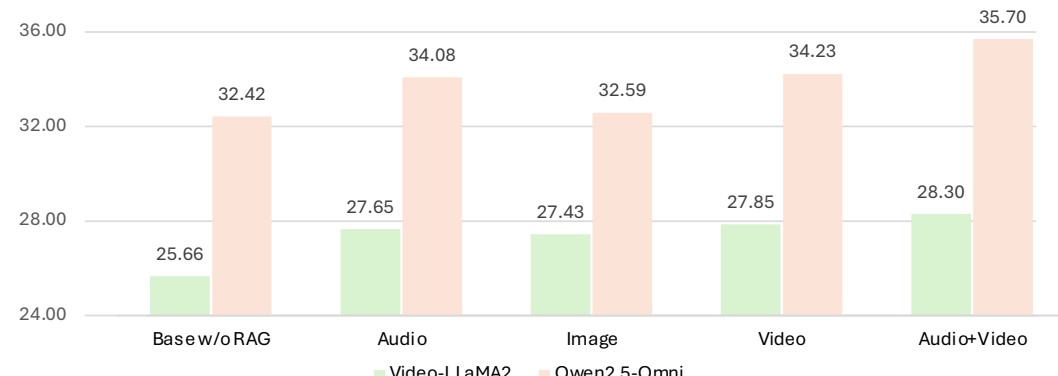

Figure 12: **Ablation study on Modality-Wise Retrieval.** We evaluate four retrieval strategies based on the modality used: audio, image, video, and audio+video.

As shown in Fig. 12, while each unimodal retrieval condition leads to improved performance over the MLLM baseline without RAG, the knowledge retrieval that considers both audio and visual modalities yields the most significant performance gain. These results highlight the importance of leveraging a wide range of modalities in MMKGs, and underscore the value of VAT-KG as a concept-centric, knowledge-intensive resource that contains rich multimodal information.

**Retrieval Checker** To demonstrate the effectiveness of our proposed Retrieval Checker, we conduct an ablation study on the retrieval checker. Tab. 7 demonstrates that the retrieval checker markedly improves performance in our multimodal RAG framework. Although retrieved knowledge already benefits the MLLM, applying the checker to filter for more query-relevant information yields additional gains across all QA tasks.

### E.2 COMMERCIAL MLLMS WITH VAT-KG

To further demonstrate that knowledge from VAT-KG also benefits advanced commercial MLLMs, we evaluate GPT-4o Hurst et al. (2024) and Gemini-2.5 Comanici et al. (2025) within our multimodal RAG framework.

As shown in Tab. 8, incorporating VAT-KG leads to consistent performance improvements across all state-of-the-art MLLMs. These findings suggest that VAT-KG is broadly effective: it enhances relatively lightweight models by supplementing missing external knowledge, and it also improves the response quality of strong commercial MLLMs, which can still suffer from fine-grained multimodal grounding despite their scale.

By providing detailed, query-specific multimodal context, VAT-KG helps close these gaps and improves overall reliability.

### E.3 DIAGNOSTIC EVALUATION ON KNOWLEDGE-INTENSIVE CASES

While our approach yields consistent performance gains across diverse multimodal benchmarks, most of these benchmarks are not designed to test knowledge-intensive scenarios. As a result, the improvements enabled by VAT-KG may appear modest. To more directly evaluate this setting, we

Table 8: **Performance on Commercial MLLMs.** Even in advanced state-of-the-art models, VAT-KG provides consistent improvements across benchmarks, highlighted in **bold.**

| Method | Knowledge Graph | Audiocaps-QA | VCGPT | VALOR |
|---|---|---|---|---|
| GPT-4o | None | 56.74 | 49.68 | 46.02 |
| GPT-4o | VAT-KG | **57.70** | **51.49** | **55.86** |
| Gemini2.5-Flash | None | 49.51 | 51.00 | 67.18 |
| Gemini2.5-Flash | VAT-KG | **50.07** | **51.12** | **69.17** |

Table 9: **Performance on Knowledge-Intensive Scenario.** Incorporating VAT-KG yields substantial performance gains on challenging QA pairs sampled from VALOR, even for commercial MLLMs.

| Method | Knowledge Graph | Knowledge-Intensive QA |
|---|---|---|
| Qwen2.5-Omni | None / VAT-KG | 26 / **46** |
| Gemini2.5-Flash | None / VAT-KG | 58 / **68** |
| GPT-4o | None / VAT-KG | 44 / **74** |

constructed a diagnostic set of 10 knowledge-intensive QA pairs featuring challenging concepts sampled from the VALOR benchmark (e.g., *tiltrotor*, *locomotive*, *electric organ*).

As shown in Tab. 9, incorporating VAT-KG produces substantial performance gains in these challenging scenarios, even for advanced commercial MLLMs. This result underscores the value of VAT-KG in enhancing multimodal reasoning under knowledge-intensive conditions.

### E.4  STATISTICAL VALIDATION OF THE MODEL AS JUDGE

To rigorously examine whether the performance gains from VAT-KG are not merely within the noise range of the M.J. scoring framework, we conducted 10 independent trials across benchmarks. Tab. 10 reports the mean M.J. scores across 10 runs, along with the standard deviation ($\sigma$) for each setting and p-value ($p$) from a t-test.

As shown in the Tab. 10, while minor fluctuations were observed, the difference remained consistently lower than the magnitude of the average performance improvements. Crucially, the mean scores consistently indicate clear performance improvements with VAT-KG, providing strong evidence that the gains are not attributable to judge noise.

To further validate this, we conducted a t-test between the baseline method and the scores with VAT-KG applied. The resulting p-value was below 0.10, confirming that the performance gains are statistically significant and not attributable to measurement noise.

### E.5  MORE QUALITATIVE RESULTS

We provide additional qualitative comparisons between the base MLLMs and our multimodal RAG framework equipped with either M²ConceptBase or VAT-KG, on downstream tasks across modalities, including AQA, VQA, and AVQA.

For AQA, Fig. 13 illustrates a qualitative comparison between the base MLLMs and our multimodal RAG framework using VAT-KG. Our framework effectively retrieves knowledge relevant to the query audio from VAT-KG and provides contextually appropriate information, guiding MLLMs toward more accurate responses.

For VQA, as illustrated in Fig. 14, MLLMs struggle to capture knowledge-intensive concepts such as "discus" or "Tai Chi". Concept-level knowledge provided by M²ConceptBase often fails to assist the MLLMs and, in some cases, introduces misleading information that exacerbates hallucination. In contrast, our framework retrieves contextually relevant knowledge from VAT-KG based on the query video, helping the MLLMs better understand and reason about complex concepts present in the scene.

Similarly, in the AVQA task, MLLMs struggle to capture knowledge-intensive concepts in the scene that also require joint understanding over both audio and visual modalities. Furthermore, image-based

Table 10: **Statistical Validation of M.J. Evaluation.** Mean scores across 10 independent trials with standard deviation ($\sigma$) and t-test p-values ($p$), showing consistent performance gains with VAT-KG beyond judge noise, highlighted in **bold.**

| Method | Knowledge Graph | Audiocaps-QA | VCGPT | AVQA | VALOR |
|---|---|---|---|---|---|
| Qwen2.5-Omni | None | 49.18 ($\sigma$=0.29) | 43.01 ($\sigma$=0.19) | 93.01 ($\sigma$=0.02) | 32.51 ($\sigma$=0.15) |
| Qwen2.5-Omni | $M^2$ConceptBase | 49.28 ($\sigma$=0.35) | 43.06 ($\sigma$=0.21) | 92.59 ($\sigma$=0.02) | 32.70 ($\sigma$=0.19) |
| Qwen2.5-Omni | VAT-KG | **50.09** ($\sigma$=0.21, $p$=1.4e-4) | **43.19** ($\sigma$=0.23, $p$=0.08) | **93.03** ($\sigma$=0.02, $p$=0.037) | **35.45** ($\sigma$=0.26, $p$=2.2e-14) |

retrieval from $M^2$ConceptBase fails to enhance the MLLMs' answers, primarily because the retrieved knowledge lacks grounding in an audio-visual context. However, knowledge retrieved from VAT-KG exhibits strong relevance to the multimodal query and helps MLLMs to better understand complex concepts such as excavator and hail, as illustrated in Fig. 15.

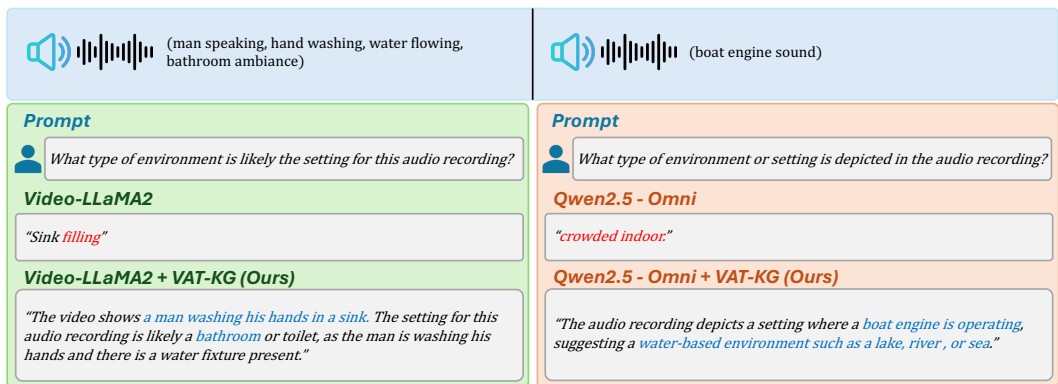

Figure 13: **Qualitative comparison on AQA task.** Incorrect responses are highlighted in red, while correct answers are shown in blue.

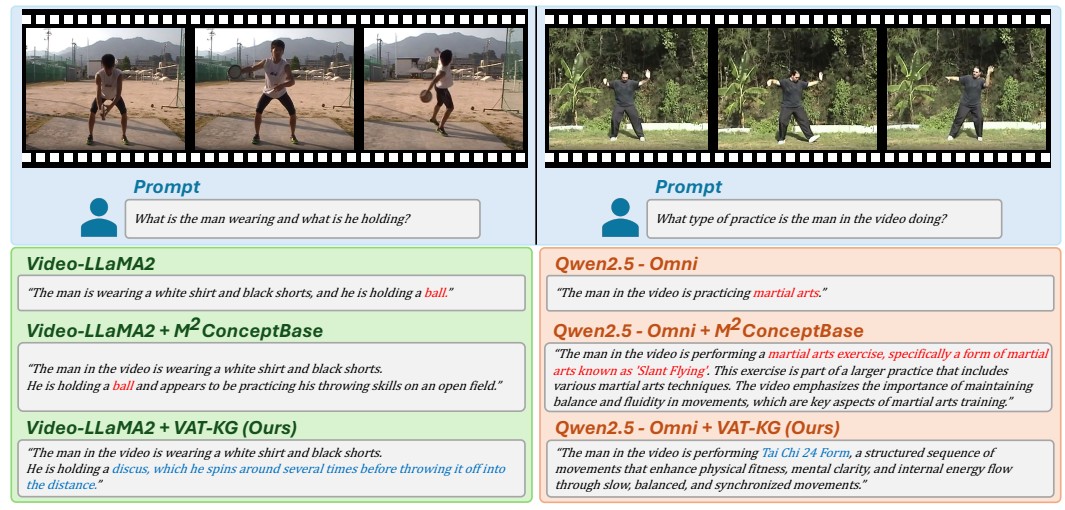

Figure 14: **Qualitative comparison on VQA task.** Incorrect responses are highlighted in red, while correct answers are shown in blue.

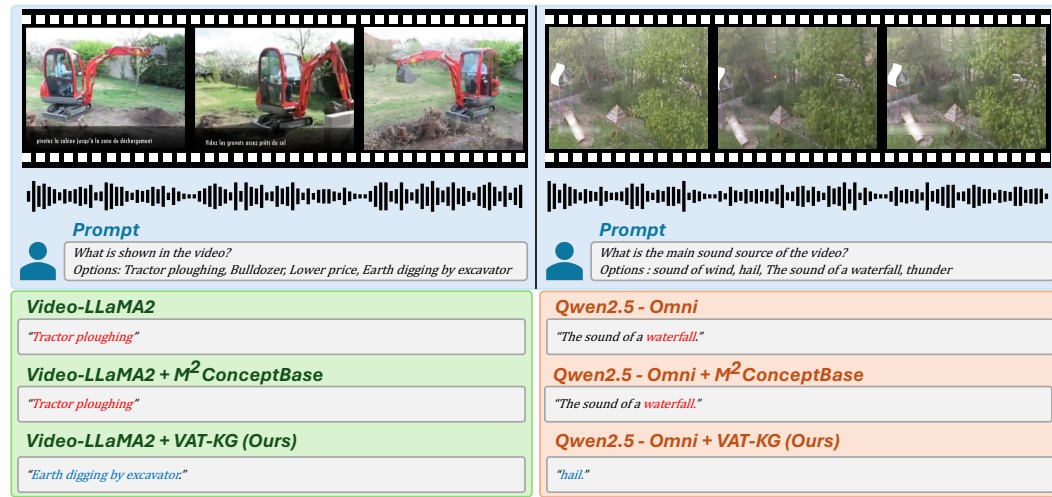

Figure 15: **Qualitative comparison on AVQA task.** Incorrect responses are highlighted in red, while correct answers are shown in blue.

## THE USE OF LARGE LANGUAGE MODELS (LLMS)

We use large language models (LLMs) solely to aid and polish writing (e.g., grammar, wording, and minor stylistic edits of author-written text). LLMs are not used for research ideation, experiment design, or substantive drafting.

