# OpenReview forum: "VAT-KG: Knowledge-Intensive Multimodal Knowledge Graph Dataset for Retrieval-Augmented Generation"
_ICLR.cc/2026/Conference — ICLR 2026 Conference Withdrawn Submission_

### Official Review · Reviewer_Bjtk · 2025-10-22

**Soundness:** 3
**Presentation:** 3
**Contribution:** 3
**Rating:** 8
**Confidence:** 4

**Summary:**

This work aims to construct a multimodal knowledge graph that (i) contains text, image, video, and audio modalities, and (ii) is aligned with text descriptions. This allows VAT-KG to reach new knowledge coverage over previous MMKGs. The authors also provide a RAG baseline that utilizes the new features introduced by VAT-KG, which demonstrates superiority experimentally.

**Strengths:**

1. VAT-KG has achieved new height of knowledge coverage.
2. The construction pipeline is carefully designed, with very rigorous filtering rules to ensure high quality of the final outcome.
3. The experiments have shown consistent improvement.

**Weaknesses:**

1. (Major) It seems that the entire construction pipeline focuses on refining the text modality, while the data of other modalities remain the same as what the initial corpus contain. Do the authors plan to have additional multimodal data crawled from the web?
2. (Minor) While this MMKG includes both image and video modalities, the paper uses "visual" to denote both of them early on. This might cause some confusion, making readers think there are only three modalities.
3. (Minor) There are two keywords in the title: knowledge intensive and RAG. In the introduction section, the first keyword is illustrated in detail, but I feel there needs to be more discussion on the second keyword. Specifically, how is VAT-KG specially designed for the RAG purpose?
4. (Minor) It appears that the authors are not using \citet and \citep correctly for in-text and parenthesis citations.

**Questions:**

See Weaknesses.

---

### Official Review · Reviewer_ha9f · 2025-10-27

**Soundness:** 2
**Presentation:** 3
**Contribution:** 2
**Rating:** 4
**Confidence:** 4

**Summary:**

This paper proposes a concept-centric visual-audio-text trimodal knowledge graph (VAT-KG). A four-stage construction process is designed, comprising multimodal alignment filtering, knowledge-intensive recaptioning, multimodal triple grounding, and cross-modal description alignment. The authors employ VAT-KG as the external knowledge base for video, audio, and audio-video understanding tasks. By incorporating modality-independent retrieval and retrieval verification, the knowledge from VAT-KG is injected into baseline models such as Video-LLaMA2 and Qwen2.5-Omni, leading to improved performance for both.

**Strengths:**

1. The motivation for building a joint visual-audio-text trimodal knowledge graph is meaningful and offers promising prospects for multimodal understanding tasks.
2. By incorporating knowledge from the constructed VAT-KG, the baseline models Video-LLaMA2 and Qwen2.5-Omni achieve modest performance improvements on video, audio, and audio-video understanding tasks.

**Weaknesses:**

1. First, from the results in Table 2, the performance improvement of the proposed method on VCG and AVQA is marginal. In fact, I believe that datasets such as AudioCaps-QA, VCGPT, and AVQA, which rely more on common sense than knowledge, are not well suited for knowledge graph applications. The authors should focus more on knowledge-related video and audio understanding datasets to better leverage the power of knowledge graphs. In addition, a comprehensive comparison with existing state-of-the-art open-source and closed-source MLLMs should be provided, rather than showing improvements only over the chosen baselines.
2. During the construction of the knowledge graph, how are entities and concepts distinguished? In Figure 2, terms like “bridge” and “helicopter” appear to be entities as well.
3. In L197, why are explanatory videos with background music removed? What is the underlying insight behind this design?
4. In L257, since the goal is to build a knowledge base for video retrieval, why are embeddings extracted from video frames rather than from segments? Segment-level representations may capture temporal semantics more effectively.
5. In L303, the phrase “the number of multimodal data per concept” is ambiguous, what exactly does “number of multimodal data” refer to in this context?
6. When performing retrieval checks, is it appropriate to match the query embedding with the text embedding of the triple? The prompt "Describe in detail..." for description tasks, as shown in Figure 6, does not seem to effectively match concepts in the knowledge graph.

**Questions:**

The reference format used in this paper seems unusual. For example, "retrieval Alberts et al. (2021)" in L37 is often written as "retrieval (Alberts et al., 2021)."
My concerns with this paper are detailed in the weaknesses section, and I would be happy to discuss them with the authors to further consider my final score.

---

### Official Review · Reviewer_fjZJ · 2025-10-29

**Soundness:** 2
**Presentation:** 3
**Contribution:** 2
**Rating:** 4
**Confidence:** 4

**Summary:**

This paper introduces VAT-KG, a knowledge-intensive, concept-centric multimodal knowledge graph that includes visual, audio, and textual modalities, specifically designed to enable RAG for MLLMs. The authors detail a rigorous four-stage pipeline for VAT-KG construction, emphasizing stringent cross-modal alignment and concept-level knowledge enrichment. The work further proposes a multimodal RAG framework that retrieves semantically aligned, detailed concept descriptions for queries from arbitrary modalities. Extensive experiments on multiple QA benchmarks demonstrate that VAT-KG consistently boosts MLLMs' performance over previous multimodal and text-only knowledge graphs.

**Strengths:**

1. The authors clearly identify a central limitation of existing MMKGs that they provide shallow or entity-centric structures and narrow modality coverage, which hinders their utility for retrieval-augmented generation with state-of-the-art MLLMs. VAT-KG’s ambition to unify video, image, audio, and text into a richly described, concept-level knowledge graph fills a visible gap and aligns with unmet needs in the community.
2. The paper introduces a comprehensive and well-justified pipeline that goes beyond simple data aggregation, while generating a dataset for MMKG.

**Weaknesses:**

1. I am very concerned about the fairness of the comparison in this article. Since the knowledge contained in VAT-KG is not universal but case-specific. As shown in the sample in Figure 2, the knowledge of an airplane flying over only appears in this picture, while existing MMKG methods are usually targeted at general knowledge retrieval scenarios. I think this comparison is not fair.
2. There have been many works on MMKG-related datasets in recent years. The characteristics of the VT-KG dataset are limited, which restricts the contribution of the paper.
3.  While the ablation on retrieval strategies in Figure 12 is insightful, the empirical analysis remains heavily oriented towards the effectiveness of integrating VAT-KG into generic QA. There is less attention paid to dissecting which aspects of the visual, audio, or video modalities most impact performance in challenging or ambiguous scenarios, or how VAT-KG’s design specifically delivers benefits for multi-modality vs. strong single-modality queries.

**Questions:**

See Weaknesses

---

### Official Review · Reviewer_G1kN · 2025-11-01

**Soundness:** 2
**Presentation:** 1
**Contribution:** 2
**Rating:** 2
**Confidence:** 4

**Summary:**

The paper introduces VAT-KG, a large-scale, concept-centric multimodal knowledge graph that text, image, audio, and video modalities. The paper proposes a four-step automated pipeline to construct VAT-KG from Video-Audio-Text Corpora, Encyclopedia and LLM, and demonstrate its use in a multimodal RAG system. Experiments on AQA, VQA, and AVQA tasks show consistent performance improvements over existing knowledge graphs such as Wikidata5M, VTKG, and M2ConceptBase.

VAT-KG is the first MMKG covering all four major perceptual modalities with concept-level descriptions. It provides a useful dataset for future research on multimodal reasoning and retrieval. However, the paper lacks a clear explanation of the specific problem that VAT-KG is designed to solve. It focuses on expanding modality coverage rather than addressing a well-defined knowledge-intensive reasoning challenge. The proposed multimodal RAG system is essentially a traditional RAG pipeline employing existing encoders (e.g., CLIP, CLAP) with VAT-KG as a new data source. No new retriever, alignment, or reasoning mechanism is introduced. In the experiments, the reported performance gains may result from diverse reasons (e.g. dataset bias or contextual enrichment) rather than genuine reasoning improvements. The paper does not analyze the underlying causes of these gains.

The dataset is an valuable contribution. However, the author aims to elevate it into a work that boasts great innovation in both data and methodology. This mismatch has resulted in a very awkward and convoluted writing logic in the paper.

Overall, this paper represents a good dataset and system contribution but is limited in methodological innovation, making it more suitable for data-centric or knowledge graph venues than for modeling-focused conferences.

**Strengths:**

1. This paper introduces VAT-KG. It is the first concept-centric multimodal knowledge graph that simultaneously covers text, image, audio, and video modalities, providing a potentially influential data resource for future multimodal reasoning research involving video and audio modalities.

2. The four-stage construction pipeline for building VAT-KG is clearly described and reproducible, demonstrating strong system integration and engineering quality. It comprises multimodal alignment filtering, knowledge-intensive recaptioning, triplet extraction, and cross-modal description alignment.

3. VAT-KG, together with the proposed RAG framework, shows significant performance improvements on audio, video, and audio-visual QA tasks, indicating the practical utility of VAT-KG’s concept-level descriptive knowledge in perception-driven scenarios.

**Weaknesses:**

Seen in Questions.

**Questions:**

1. VAT-KG

1.1 Motivation and Novelty of VAT-KG

The contribution of VAT-KG lies in engineering integration rather than in a cross-modal semantic breakthrough. Modality diversity serves more as a narrative hook than as a justified research necessity. The paper does not clearly explain why constructing such a knowledge graph covering 4 modalities is scientifically or methodologically important.

As shown in Figure 1, the contrast among (a) non-concept-centric, (b) concept-centric with limited modalities, and (c) the proposed VAT-KG only shows that no one has done this combination before, which alone does not constitute a research challenge.

The true challenge should stem from non-trivial difficulties encountered when extending concept-centric MMKGs beyond image–text pairs, such as semantic granularity mismatches across modalities, one-to-many or many-to-one mapping conflicts, or other issues introduced by incorporating new modalities.
If the motivation were reframed around the requirements of specific knowledge-intensive downstream tasks and the alignment challenges introduced by the mere engineering integration of modalities, rather than simple modality expansion, then VAT-KG would present itself as a genuinely challenging and meaningful contribution.

1.2 Why are relationships in VAT-KG based on co-occurrence rather than knowledge-intensive concept-level semantics?

The example triplet in Figure 2 ("helicopter flies over bridge") reflects co-occurrence rather than knowledge-intensive concept-level semantics. The visual and audio modalities simply co-occur, without showing clear reasoning or interaction between them. Thus, this example underplays VAT-KG’s claimed conceptual strength.

A more convincing example would show how different modalities work together. For instance, how sounds match events or how visual and audio cues help reasoning. Otherwise, the proposed VAT-KG is only a dataset / knowledge base, not a graph.

2. The proposed RAG Framework

In this paper, the proposez multimodal RAG framework simply applies existing multimodal encoders to retrieve concept descriptions from VAT-KG and feeds these descriptions into an MLLM for generation. Thus, the so-called framework is more as a standard RAG pipeline operating on a new multimodal knowledge source, rather than as a newly improved RAG method. It introduces no new architectural design or reasoning component: no new retriever structure, no new cross-modal encoder, no new fusion or interaction mechanism, and no new retrieval scoring or reranking strategy.

A innovative multimodal RAG need to address new challenges specific to VAT-KG, such as modality alignment issues, granularity mismatches, or other cross-modal inconsistencies introduced by additional modalities.

3. Soundness of Experimental Results

The claims about VAT-KG’s experimental performance improvements are overstated and require clearer justification. The experiments validate VAT-KG’s utility as a multimodal retrieval source but do not establish its necessity for addressing knowledge-intensive multimodal reasoning or mitigating hallucination (mentioned in line 472).

3.1 Overstated: Downstream Task

The reported gains mainly appear on perception-oriented tasks (AQA, VQA, AVQA), such as “What instrument is playing?” or “What sound occurs?”, where the improvement may be from the closer modality match between VAT-KG and the benchmarks. VAT-KG’s closer modalities match to the benchmarks. The authors should clarify whether the performance gains come from complementary multimodal information or simply from modality alignment. If it is the latter, a multimodal knowledge base would suffice, rather than a knowledge graph, and therefore relation extraction is not needed.

3.2 Unfair Comparison between VAT-KG and other MMKGs

The comparison against Wikidata5M, VTKG, and M2ConceptBase is inherently biased, as these baselines lack audio/video modalities, giving VAT-KG a built-in advantage in video/audio tasks.
Thus, the experiments effectively measure modality matching rather than knowledge effectiveness.

3.3 Missing Convincing Explanation of Improvements

The logic in this paper is circular: you build a new KG covering video and audio along with descriptions ${\rightarrow}$ use it in RAG  ${\rightarrow}$  outperform other KGs without video or audio  ${\rightarrow}$  therefore it works. The logic seems to be incorrect.
This paper doesn't explain why other MMKGs perform poorly or why VAT-KG performs better except the only expaned modalities resouces.
To strengthen the argument, the authors should identify whether the gains are from inclusion of extra modalities or better knowledge structure and alignment of VAT-KG.

4. Writing Issues

4.1 Overclaim

The claim about the generality of its construction pipeline is overstated.

For example, Lines 93–95:

> "We introduce an effective pipeline for constructing multimodal knowledge graphs from arbitrary multimodal corpora…"

The proposed pipeline is only validated on video, audio, and text modalities, without evidence of generalizability to other types such as sensor data, 3D information, or gesture-based modalities.

Therefore, the pipeline should be described as a modality-specific integration system, not a universal multimodal construction framework.

4.2 Logical Inconsistency

Line 15-19:

> "However, existing MMKGs are generally limited in scope: they are often constructed by augmenting pre-existing knowledge graphs, which restricts their knowledge, resulting in outdated or incomplete knowledge coverage, and they often support only a narrow range of modalities, such as text and visual information."

The discussion of prior MMKG limitations conflats two distinct issues:

1\) Outdated knowledge caused by a lack of timely updates and maintenance.

2\) Limited modality coverage because existing KGs only support text–image data.

These are independent problems with different causes and implications, yet the paper presents them as a single causal chain.
Extending to new modalities does not inherently address the problem of outdated or incomplete knowledge.
This paper should clarify which limitation VAT-KG primarily aims to resolve.

5. Suggestions for Revision

5.1 Motivation Should be Clear

The paper’s current motivation is not clear: Existing MMKGs lack modalities ${\rightarrow}$  We build a more complete MMKG ${\rightarrow}$  It improves downstream tasks involving these modalities. This coverage-driven narrative makes the work appear as a dataset extension rather than a solution to a concrete knowledge-intensive problem.

A clearer motivation would show when perception alone fails and world knowledge is needed. VAT-KG should then be presented as a structured bridge to close this gap.

5.2 Reframing Based on Figure 6 (a very good example)

The insight of the paper lies not in its multimodal coverage but in the knowledge-guided perception phenomenon illustrated in Figure 6.
This example shows that MLLMs often misinterpret perceptual inputs due to the lack of video conceptual knowledge, while VAT-KG provides structured knowledge that corrects these perceptual errors (e.g., identifying a "synthesizer" rather than a "keyboard").

Reframing the paper around this problem would transform VAT-KG from a data-engineering contribution into a substantive study of knowledge-intensive multimodal reasoning. A data-engineering contribution is more suited for venues emphasizing knowledge graph construction, data systems, or multimodal data management (e.g., ICDE, CIKM) .

---

### Note · Authors · 2025-11-16

I have read and agree with the venue's withdrawal policy on behalf of myself and my co-authors.